# The role of safety in modal choice and shift: A transport expert perspective in the state of Victoria (Australia)

**Mohammad Nabil Ibrahim**⊙\*, **David B. Logan**⊙, **Sjaan Koppel**⊙, **Brian Fildes**

Monash University Accident Research Centre, Monash University, Melbourne, Victoria, Australia

\* mohammad.ibrahim@monash.edu

**Data Availability Statement:** All relevant data are within the paper and its Supporting Information files.

**Funding:** The author(s) received no specific funding for this work.

## Abstract

### Background

Previous research showed differences in the exposure to risk from using different modes of transport and that modal choice can significantly impact road safety outcomes. Though, a modal shift to a safer mode is not commonly discussed as part of road safety strategies.

### Aim

This study aimed to explore the perspectives of transport policymakers about the role of safety in modal choice and if it can be one of the main factors for modal choice and shift.

### Method

Seven semi-structured interviews were conducted with transport experts from government (n = 5) and private (n = 2) organisations in the state of Victoria. Interview transcripts were analysed using a thematic approach to identify the key perspectives of the experts.

### Results

Overall, the analyses indicated uncertainty of the ability to use safety in modal choice as a road safety strategy and identified two main issues; 1) the perceived limited role that safety plays in people's modal choice, and 2) that safety is perceived to be a barrier to modal choice and modal shift towards public and active travel. Experts suggested that when considering transport modes other factors such as convenience, availability, speed, cost, trip purpose and income are more influential than safety in modal choice. They also suggested that safety might play a role within the chosen mode, but not in choosing between modes, such as considering safety features when purchasing a car after deciding to drive a car. It was also stated that safety could act as a barrier preventing people from choosing sustainable transport modes of public transport and active travel.

### Conclusions

Theoretically, it is argued that safety and mobility cannot be traded against each other, and that mobility becomes a function of safety, not vice-versa. However, our findings indicated

**Competing interests:** The authors have declared that no competing interests exist.

that the transport experts did not believe that safety is the main factor in the modal choice process. Transport experts believed users choose their mode of transport mainly to achieve mobility benefits without necessarily considering how safe is their choice as a differentiator factor. While the shift to a safer mode of transport would help improve road safety outcomes, further investigations are needed to inform how can we influence the consideration of safety as the main factor in modal choice and removing barriers to using the relatively safest available mode of transport.

## Introduction

Travel and transportation options have been continually evolving, and with every technological evolution, different modes of transport have been introduced with different consequences. Despite the benefits of transportation to accessing work, education or other community and social activities, there are also negative consequences, including crash-related deaths and serious injuries, traffic congestion, air pollution, and noise [1]. Tingvall and Haworth [2] argued that safety is the most important factor in the road transport system, with the loss of human life and/or health unacceptable to society, and therefore the system must be designed such that crash-related deaths and serious injuries are eliminated. Mobility, therefore, should be secondary to safety and should not be obtained at the expense of safety.

### The safe system approach

Australia was one of the first countries to formally adopt the Safe System approach to improve road safety [3]. The objective of the Safe System approach is that eventually no one will be killed or seriously injured within the road transport system. This has been embraced in a "Towards Zero" philosophy in Australia [4]. Both the Safe System approach and Towards Zero have been adapted from the Swedish "Vision Zero" initiative with a similar philosophy that was legalised in Sweden in 1997 [2]. Vision Zero provides a focus on a safe road transport system which can be used to guide the selection of strategies and then the setting of goals and targets. Vision Zero focuses on the way the road transport system should be designed to ensure zero deaths and serious injuries. Tingvall and Haworth [2] argued that safety and mobility cannot be traded against each other, and that mobility becomes a function of safety, not vice-versa. In addition to the Safe System approach, which focuses on the road transport system, some literature identified modal choice and modal shift as a potential intervention measures to shift travellers toward modes with lower fatality and serious injury outcomes such as trains and buses [5–8].

### Modal shift

Modal shift is generally discussed to alleviate congestion and emission by reducing the number of private cars and shift users to public trains, trams or buses, and to achieve health benefits associated with active travel modes (i.e., increasing walking and cycling). Although, conceivably it could also achieve further safety benefits for the road transport system. Litman and Fitzroy [7] argued that it is important to include the concept of modal shift as part of any road safety strategy. However, very few attempts have been made to estimate the road safety benefits of applying modal shift strategies [5].

Previous research has reported that decisions about modal choice are made according to values that vary amongst individuals. Modal choice values can be related to the characters of the individual, their social and/or economic status, the journey, and/or the transport facility or vehicle [9, 10]. The modal choice decision is the outcome of individual trade-offs between different factors [11]. Safety perceptions are not always included in modelling transportation choices [12]. Many of the previous studies have discussed modal choice factors from the individual perspective, whereas some of those factors are outside the control of individual users and in the realm of transport planners and providers.

## The impact of modal choice on road safety outcomes

There are differences between the risk of different modes at different times of the day or days of the week, and that road safety outcomes are sensitive to the travel mode. The International Transport Forum collected mobility and road safety data from 31 cities to evaluate, monitor and benchmark road safety outcomes. Analysis reveals considerable differences in road safety performance between modes within the same city and between cities. Pedestrians, cyclists and motorcyclists, together called vulnerable road users, make up about eight out of ten road users killed in city traffic [13].

Furthermore, for cyclists injured in in France, the risk of being killed, both based on time spent traveling and the number of trips, was about 1.5 times higher than for car occupants [14]. Cairney [15] further claimed that cycling and walking have the highest risk of road death and serious injuries. Litman [16] noted that walking and cycling have relatively high crash rates per mile, but per-capita crashes tend to decline with increased use of these modes.

Bouaoun et al. [14] showed the risk of being killed was 20 to 32 times higher for motorized two-wheeler users than for car occupants. When compared with the U.S., motorcyclists had a fatality rate per passenger mile 29 times that for automobiles and light trucks [17]. In Australia, travel by motorcycle is by far the least safe mode of travel, with fatality and serious injury rates approximately 30 times those for travel by car [15].

Taxis and ridesharing services may reduce traffic crashes in some situations, such reducing drunk driving. Uber [18] and Litman [16] (claimed that increasing trips by competitive ridesharing and late-night transit services around bar closing times increased travel as car passengers rather than drivers, and therefore contributed to reducing the number of drunk drivers and the subsequent probability of crashes.

Bus travel has a low risk and switching from travelling by car to travel as a bus passenger is likely to have a neutral or mildly positive effect [15].

Ibrahim et al. [19] estimated the relative risk across nine travel modes in the Australian state of Victoria and found public transport modes to be the safest way of travel, while motorcycle and bicycle are the most at-risk modes.

Therefore, modal choice of different modes of travel has the potential to cause both positive and negative outcomes, depending on the type and extent of the changes and differences in road user exposure to collision severity [15].

While generally the 'Safe System' approach is accepted as the best current approach for improving road safety [20], other supplementary approaches such as management strategies aimed at shifting travel towards a more favourable and safer mode of travel can also help.

Differences in risk between different modes of travel encouraged the need to include modal shift as part of any road safety strategy in order to be able to further reduce trends in causalities and crashes [5, 13, 15, 21, 22].

## Study objectives

Merging the objectives of mobility management and safety has the potential to achieve combined benefits and lead to safer and better transport outcomes overall. The rationale behind this study was the hypothetical believe that safety comes first, and it should drive modal choice.

This study aimed to explore to what extend safety consideration is prioritised over mobility in modal choice from transportation providers and policy makers perspective to answer the main question related to the role of safety in modal choice and if safety can be used as the main factor for modal choice and shift.

## Method

A qualitative research approach was undertaken, which included semi-structured interviews with experienced transport experts from public and private transport organisations. This semi-structured approach was selected so that the researcher could adopt questions that are both fixed and open-ended and may not be asked in the same way or order for each respondent. This allowed the participants to elaborate on any specific topics of interest and/or concern to them. A thematic analysis was used to structure the data outcomes, following previous guidelines of Clarke and Braun [23]. This method was applied to identify patterns and to explain the participants conceptualisation of the research questions without being tied to a particular theoretical viewpoint [23, 24].

Ritchie et al., [25] noted different ways to identify potential participants such as using existing lists of registered participants or by generating specific lists with specific requirements for a research study.

A purposive (selective) sample was identified for this study that enables detailed exploration and understanding of the perspective of transport organizations towards modal choice and safety. Setting a selective criterion for the participants enhance how potential group differences and similarities can be illuminated to address the subject under study and the research questions [24–26].

The following criteria were set to short list potential participants:

- Participants are employed by a public or private transport organisation, and

- participants are responsible for transport planning and/or operation in their organisation.

A list of potential participants was identified that included 16 stakeholders: six from state agencies, three from transport operators, four from private organisations, two from local councils, and one from academia.

### Participant selection

Recruitment was undertaken according to Monash University's ethical guidelines. Contact with potential participants was made via email which included an invitation to participate, an outline of the research under discussion and a consent form, allowing the participants to make a more informed decision about whether to accept or decline the invitation. The response rate to the 16 invitations sent out was 44 percent (n = 7).

Although, the sample size is relatively low but for detailed interview with experts, this number is adequate as suggested by previous qualitative researchers. Even samples as small as four or five is enough to make valid comparisons and can be very effective in revealing group perceptions rather than individual perceptions [24–26].

Once each participant agreed to participate in the study, an interview was organised using either face-to-face, telephone and/or video interviews, depending on their suitability. Six of the seven interviews were recorded to aid accurate transcription; however, one participant did not agree to the recording and hence only notes were taken. Details of the sample are shown in Table 1 below.

### Interview questions

Fourteen structured interview questions were prepared for each interview and listed in Table 2 below. The interview questions were developed based on the findings of previous research studies, as well as to address the research objectives. The interview started with general questions about transport problems, leading to more focused questions about modal shift and travel safety. As noted above, due to the semi-structured nature of the interview, each participant was encouraged to elaborate on each question where they wanted to.

### Procedure

The semi-structured interviews were designed to take approximately 30 minutes to complete due to the busy schedule of the participants. However, the average interview time was 45 minutes (due to the additional time needed to expand the discussions from the initial questions).

Following the completion of the interviews, all the raw data (including recordings and notes) were processed and transcribed in preparation for the analysis using the Thematic Analysis approach.

The Thematic Analysis depends on interpretation including coding, categorisation, and noting patterns [27]. Joffe and Yardley [28] also noted that the theme must also describes the bulk of the data. In this study, an inductive approach of descriptive coding was followed and simultaneous coding was used when the data's content suggested multiple meanings [29]. Each code was described using clear operational definitions so they can be applied consistently by a single researcher over time, and multiple researchers can use the same definitions as they code future data [29].

### Results

The themes and patterns that emerged from the analyses of the participant responses are shown in Table 3 below.

The following two main themes were identified and are discussed further below.

- Perceived limited role of safety in modal choice of transport, and

- The role of safety as a barrier to modal choice and modal shift

#### The role of safety in modal choice

The experts believed that modal choice is rational and the role of transport safety in modal choice is limited, and the following factors are related to this perception.

Trust was perceived as a strong reason to minimise safety concerns regarding modal choice. Trust can be divided into three categories: 1) trust in the transport system, 2) personal self-ability trust, and 3) organisational self-ability trust.

Trust in the system was noted in the response of Participant 3 to Q3: "It is not clear if modal choice can be used as a safety strategy because the expectation from transport users is

**Table 1. Details of the stakeholders who participated in the interviews.**

| | Organisation | Organisation/Participant Role | Interview format | Duration (mins) |
|---|---|---|---|---|
| 1 | Public Transport Victoria (PTV) | Planning for public transport network development including train, tram and bus services | Face-to-Face | 42 |
| 2 | Bus Association | Represent and advocate the public transport industry | Face-to-Face | 58 |
| 3 | Royal Automobile Club of VIC (RACV) | Addressing community needs in the areas of mobility | Face-to-Face | 35 |
| 4 | Transport Safety Victoria (TSV) | Manage transport safety standards including legislation, licensing, registering and monitoring | Face-to-Face | 50 |
| 5 | Transport Accident Commission (TAC) | Promote road safety, improve the State's trauma system and support those who have been injured in road crashes | Telephone Interview | 50 |
| 6 | POPCAR Car Sharing | Car rental club for short-term car rental | Video Call | 40 |
| 7 | Transport for Victoria | Transport Planning | Video Call | 58 |

that all modes are safe and the transport system that is provided by the government should be safe".

Trust in personal ability was highlighted in response to Q6 about the role of safety in modal choice, Participant 2 stated: "…people who drive do not think that they are going to have accidents, but it happens, and it could be not your fault–people do not think about that."

Similarly, trust in the organization was coded with Participant 6 stated that his organization trusted the safety of the mode they provide: "our car is new, and the age of our fleet is newer than typical private vehicle fleet, so it is somehow safe in terms of the quality, technology and features of the vehicles". and he stated that as they have new fleets "technology and features of cars is a safety advantage".

Within Mode was used by participants explicitly stating that safety should be addressed within the chosen preferred mode but not to influence travel choice. Participant 3 believed their role was: "telling people how they can be safe in a particular mode but not telling people to choose a particular mode". Participant 3 stated when asked Q10: "No, I don't believe that getting people to choose safer modes of travel would be an effective solution; if you are choosing between one mode and another it is not realistic to base your decision on safety only, but if you are choosing within the same mode, then it is definitely a factor such as choosing to

**Table 2. The semi-structured interview questions.**

| | |
|---|---|
| **Q1** | **What do you think are the major problems with the Victorian transportation system at present?** |
| **Q2** | What are the possible solutions under consideration? |
| **Q3** | If we are looking for Modal Choice to improve safety, can you explain what is meant by Modal Choice as a potential strategy for improving road safety? |
| **Q4** | Do you have any thoughts or policies for using modal choice to improve road safety? |
| **Q5** | What do you believe are the factors that influence the way people choose their travel modes? |
| **Q6** | What role do you think safety plays in these choices? |
| **Q7** | What do you see as your organization role in influencing people's choice of mode of travel? |
| **Q8** | Does your organization actively encourage people to choose safer modes of travel? |
| **Q9** | How–policies, practices, etc.? |
| **Q10** | Do you think that getting people to choose safer modes of travel would be an effective solution? |
| **Q11** | How would modal choice fit into a safe system approach to road safety? |
| **Q12** | How do you see the role of car sharing or ride sharing (GoGet, Flexidrive, Uber) on influencing travel choices? |
| **Q13** | What do you believe is possible impact of disruptive technology in the way people choose to travel? |
| **Q14** | Other thoughts or suggestions? |

Table 3. Organising themes, codes and main themes related to modal choice shift and travel safety.

| Organising Themes | Codes | Main Themes |
|---|---|---|
| **Mode Factors** | Trust | Limited Role for Safety in Modal Choice |
| | Within mode | |
| | Convenience | |
| | Availability | |
| | Speed | |
| | Cost | |
| | The behaviour of other users | Safety as Barrier to Modal Choice and Shift |
| | Integration | |
| | Awareness/Information | |
| **Trip Factors** | Trip Purpose | Limited Role for Safety in Modal Choice |
| | Night-Travel | Safety as Barrier to Modal Choice |
| **Individual Factors** | Income | Limited Role for Safety in Modal Choice |
| | Time availability | |
| | Age | Safety as Barrier to Modal Choice and Shift |
| | Gender | |

purchase a different make of car". Similarly, Participant 7 stated that after people decide to drive a car for factors other than safety, safety might play a role in which car to drive: "when people decide to purchase a new car".

Non-Safety factors those that don't include safety specifically dominated modal choice responses. For example, Participant 1 noted that: "If you try to influence people's choice from safety only it will be a very weak strategy", and Participant 2 noted: "I don't think safety will influence their decision". The following factors were perceived to be important in modal choice which are not necessarily related to safety including:

- Convenience: this factor included the perceived convenience of the mode. For example, in response to Q5 about the factors that influence the way people choose their travel modes, Participants 1, 4, 6 and 7 explicitly stated that 'convenience' was one of the most important factors. Participant 7 stated that "convenience is the reason why people in the outer and regional areas prefer cars". We also referred to convenience from participant's notes as "ability to move smoothly"," internal design of the buses"," enhancement to physical on and off the bus"," frequent and direct bus service", "availability of bike racks" and "integration between different modes".

- Availability: alternative's availability was highlighted as one of the main modal choice factors, as stated by Participant 3 in response to Q5:" the availability of the modes and transport options".

- Speed: It was referred to as a main factor in modal choice as per responses to Q5 such as stated by Participant 1 describing why the car is more attractive: "public transport usually is slower than private vehicle". Participant 3 also stated that "travel time can be related to the mode speed".

- Cost and Income: The cost of travel was also referred to as an important factor. Participant 1 stated that people drive more due to "the perception of the cost of driving is quite a low and people don't understand clearly the cost of their driving". In addition, Participant 6 believed that people choose the mode that "works financially for them". In discussing the organisation's role in influencing people's choice of mode of travel (i.e., Q7), Participant 1 referred to

the income level and stated that "students are primary users of public transport due to low car ownership and income".

- Trip purpose: Participant 2 mentioned "the purpose of the trip"in response to Q5 as a main role in the modal choice.

- Time availability: Participant 2 responded to Q5 by linking modal choice to "how much is the time availability" for different individuals and their specific trips.

## Safety as a barrier to the modal choice and shift

Safety was perceived as a barrier to public transport and active travel and that can result from the following factors:

- The behaviour of others: For example, Participant 4 noted in response to Q2: "bus drivers to drive more carefully and gently, other road users to respect bus movements and consider people inside the buses". In addition, Participant 1 responded to Q8 that: "customer behaviour within public transport" might be a personal safety concern for some people that prevent them from choosing safe modes such as public transport.

- Road user age: The issues of undesired behaviour from other users as stated above highlighted the age barrier of elderly people in using buses and that age might play a role in modal choice when they have specific concerns. Age was also discussed with Participant 1 regarding Q8 and the concerns of families travelling with young children when using public transport: "try to address safety issues within public transport service such organizing campaigns for the public on how to access and park a pram".

- Integration: integration between modes, especially during modes change, was highlighted by most of the participants as an issue for public transport users. Participant 3 responded to Q4 on suggesting safety policies: "to make the transport hub where people change the mode to another is more friendly and safe, particularly for pedestrian and cyclist". It was noted by Participant 4 as one of the major problems with the Victorian Transportation system at present replying to Q1: "integration of transport services especially bus service ". Participant 5 also stated that: "Enhancement to physical on and off for public buses is required to achieve better perspective toward bus safety". Participant 1 stated: "Public transport is poor in some areas and people can't get there easily".

- Awareness/Information: Participant 2 highlighted the importance of information availability to increase confidence when using any mode of transport and in busses particularly: "there is weakness in information reliability". Participant 1 also highlighted the importance of awareness campaigns on how to use public transport to address safety issues of slips and falls.

- Night Travel. Participant 2 highlighted personal security when travelling at night: "because we all were told that safety is a concern when travelling at night in public transport, personal safety is the safety concern from individual perception".

- Gender: Participant 1 highlighted the concern of travelling at night for female travellers. They also highlighted the concerns for women in cycling. In response to Q6, they stated that "there is a clear difference between men and women in cycling especially" and "Personal safety or security plays a major role, especially for women and night-time".

## Summary of results

Overall, the experts suggested that the role of considering safety in modal choice is limited, it can improve road safety when people consider safety after they choose their mode regardless of how it performs against other modes such as considering safety features when purchasing a car after deciding to drive a car. Sometimes considering safety might lead to choosing a less safe modes, such as people trusting their car is safe because of technology for example. Moreover, safety might act as a barrier to choosing safer modes of public transport because of some potential barriers such as the behaviour of others, multiple journey integration, lack of information, travel at night, road user age, and gender.

In the modal choice process, safety comes after some more important factors identified by the experts that could be further investigated and addressed to encourage the use of safer modes. Those factors as discussed above include convenience, availability, speed, cost, trip purpose and income.

## Discussion

The literature highlights that research on road safety and research on modal choice are being discussed separately. Road safety research focused on improving the road user safety of the chosen mode such as how to improve motorcyclist safety if they choose motorcycle as a mode of transport. Modal choice research focused mainly on how people choose their mode of transport from social and economic factors and on modal shift for sustainability targets. This study combined both road safety and modal choice to investigate if safety can be a factor in encouraging the choice of safer mode of transport. This study also examined the perspectives of transport providers and policymakers on what they think the transport user places a value for safety in modal choice.

This study utilized a qualitative semi-structured interview method to enable exploring a new approach to achieve road safety through modal choice and modal shift.

The sample of experts interviewed for this study represented many years of practical experience working in the area of road safety and transport planning in the state of Victoria.

While the study aimed to discover how safety can influence modal choice, the transport experts did not think safety plays a main role in people choice. There was an overall believe that people place more value to factors such as convenience, availability, speed, cost, availability of information, trip purpose, time of travel, age, gender and income over the safety of the mode relative to other modes and those factors are consistent with previous research as main factors for modal choice [30–35]. This aligns with the previous studies on modal choice such as Batty, Palacin [35] who noted that modal split has remained relatively stable over recent years as social factors and economic barriers have acted to prevent modal shift and choice.

Therefore, as transport policymakers they did not believe modal shift driven by safety is a feasible strategy. Though, evidence from previous studies showed the relatively lower risk of road injury when using public transport compared to other modes [19].

Conversely, safety when perceived as personal security might lead to shift away from public transport towards less safe options. Concerns related to the behaviour of other users were raised especially for elderly people which was discussed in the literature along with other issues related to safety concerns when getting on/off, during the journey and personal security [36, 37]. This barrier was discussed in the literature for young people also as discussed by Currie and colleagues [38] on the perceptions of personal safety on public transport. This result highlighted the need to discuss modal shift strategies based on users' demography of age group and also the need to understand the meaning of safety concerns that were discussed outside the risk of crash injury and was more about personal safety during travel.

Similar results in relation to personal safety was suggested by previous work by Alonso, Useche [39] to study the relationships between the perceived security and travel behaviour. The results of this study suggested that perceived safety, in both urban environments and public transport systems, is a relevant issue affecting the daily transport-related patterns and behavioural choices of the Dominican Republic's population. A previous study also by Delbosc and Currie [40] concluded that fears about crime-related personal safety on public transport can have an important impact on ridership.

Other barriers to use public transport were related to the mode and services such as stops and transfer points between public transport modes and the availability of information and updates. These factors were discussed in the literature along with other factors related to what influence modal choice and explain why people choose a private cars over public transport [34].

Although, the transport experts did not believe road safety could motivate modal choice of safe options, however, they believed it could discourage cycling because it is perceived as high-risk mode of road injuries. Gender was identified as a safety barrier for female cycling. This finding is consistent with that of Twaddle and colleagues [41] who demonstrated that women are less likely than men to be cyclists and suggested that if women's cycling needs were addressed, the modal share of bicycle commuting could be increased.

In addition to the social and economic factors that dominate people modal choice, the transport experts mentioned trust as a factor that can overcome thinking of safety. Trust was perceived as a strong reason to minimise safety concerns in the modal choice. People ignore other safer modes because they trust their current choice. Previous research by Armstrong and Mok [42] defined trust as what is shown by a person who has a belief that the journey to the destination is reliable about the quality of service and safety during travel.

Trust was linked to technology and the safety of new transport alternatives could play a role in minimising the safety concern and it has been discussed in the literature for the acceptance of autonomous driving [43].

Previous studies suggested that having relatively high levels of trust in others could increase preferences for public transportation and carpooling [10, 44] and the use of park and ride facilities [45].

On the other hand, trust also might be a barrier to modal choice as reported by Garrard [46] that trust in others (for both personal and traffic safety) plays a main barrier to children's active travel to school and independent mobility. Trust also might be a barrier for children to travel in current or future car-sharing modes [47].

Nevertheless, it was believed that people consider safety within the chosen preferred mode. This was highlighted in previous research related to the car purchasing process by Koppel et al. [48] demonstrated that consumers ranked safety-related factors as more important in the new vehicle purchase process than other vehicle factors (e.g., price, reliability etc.). Another example of considering safety within the chosen mode is whether to wear a helmet or not when cycling [49, 50] and motorcycling [51, 52]. Another example from the literature related to when and where to cycle specifically the safety aspects of riding with children as discussed by Hatfield and colleagues [53].

Those cases where people considered safety in their travel choices, have adequate communication materials that provide useful information for transport user to make informed decision. Availability of information was one of the identified factors by the experts that play main role in modal choice, therefore, communication campaigns can be an effective tool to encourage modal choice of the safest available mode if it the safety level communicated clearly to the transport users. This opportunity was investigated by Faus, Alonso [54] and suggested that traffic and road safety advertisements have a certain positive effect and their effectiveness is substantially increased if they are accompanied by other preventive measures such as

legislation or road safety education. Another study by Zatoński and Herbeć [55] examined whether mass media campaigns are helpful in preventing alcohol-impaired driving and found that with a focus on positive messages, mass-media campaigns can successfully contribute to improve road safety outcomes.

The results of this study highlighted the complexity of including safety as factor that can influence modal choice as suggested by the transport experts. The Safe System approach philosophy implicitly represents the experts' view, they believe that no mode shall be accepted as risky mode, safety should be a mandatory factor of every mode of transport and transport users should be encouraged to be as safe as possible within the mode they choose that meets their social and economic needs. However, in reality, not all modes have the same safety level of injury outcomes [19].

## Strengths and limitations

This is the first study to our knowledge that has attempted to gain a detailed in-depth understanding of the perspectives of transportation organisations on using modal shift to improve road safety and the factors and barriers behind these opinions. These findings are important if there is a desire for government and transportation administrators to initiate safety improvements through modal choice. The findings identified earlier are critical and in need of consideration and action to gain the best safety improvement outcome and address the barriers likely to impede the potential benefits, identified in previous research.

This study was limited by the sample size of the interviews conducted and by the questions asked, plus it focused only on exploring the stakeholder groups' concerns found within the state of Victoria, which may have been different elsewhere with the different transport systems and demographic characters. There is a need for more data on the willingness of transport users to choose their transportation modes, based on lowering their safety risk without imposing undue restrictions on their ability to travel.

## Further research

As noted above, the results of this study identified transportation administrators' perspectives. Importantly, though, the users themselves will also have strong views of the factors and barriers for them in choosing their transportation options which may or may not match the administrator's views. Such a study of individuals' views will help illustrate differences between the transport users and the policymakers in terms of motivation and importance. As there are large numbers of users, the sample size will need to be greater, using traditional survey techniques to ensure sufficient validity. This would be a useful contribution to appreciate the challenges to optimise safety through modal shift.

## Conclusions

Theoretically, it is argued that safety and mobility cannot be traded against each other, and that mobility becomes a function of safety, not vice-versa. However, our findings indicated that the transport experts did not believe that safety is the main factor in the modal choice process. Transport experts believed users choose their mode of transport mainly to achieve mobility benefits without necessarily considering how safe is their choice as a differentiator factor.

Further investigations are needed to inform how can we influence the consideration of safety as the main factor in modal choice and removing barriers to using the relatively safest available mode of transport. The results also identified different factors for modal choice that are outside the safety consideration, but it could lead to choosing less safe modes.

This study was limited by the questions asked, focused only on exploring the concerns of stakeholders within the state of Victoria, which may have been different elsewhere with the different transport systems and demographic characteristics.

It is helpful if these findings from the experts could be compared with those of individual transport users to investigate any differences between the transport users and the policymakers in terms of motivation and importance. This would be a useful contribution to appreciate the challenges in promoting a safety modal shift in Victoria.

## Supporting information

**S1 File.**
(PDF)

## Author Contributions

**Conceptualization:** Mohammad Nabil Ibrahim.

**Data curation:** Mohammad Nabil Ibrahim.

**Formal analysis:** Mohammad Nabil Ibrahim.

**Investigation:** Mohammad Nabil Ibrahim.

**Methodology:** Mohammad Nabil Ibrahim.

**Project administration:** Mohammad Nabil Ibrahim.

**Resources:** Mohammad Nabil Ibrahim.

**Supervision:** David B. Logan, Sjaan Koppel, Brian Fildes.

**Writing – original draft:** Mohammad Nabil Ibrahim.

**Writing – review & editing:** David B. Logan, Sjaan Koppel, Brian Fildes.

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
