## [Decision Letter · Decision Letter 0]

7 Oct 2022

PONE-D-22-23849EXPERT PERSPECTIVES OF MODAL SHIFT AND SAFETY IN THE STATE OF VICTORIA (AUSTRALIA)PLOS ONE

Dear Dr. Ibrahim,

Thank you for submitting your manuscript to PLOS ONE. After careful consideration, we feel that it has merit but does not fully meet PLOS ONE’s publication criteria as it currently stands. Therefore, we invite you to submit a revised version of the manuscript that addresses the points raised during the review process.

We look forward to receiving your revised manuscript.

Kind regards,

Iman Aghayan, Ph.D.

Academic Editor

PLOS ONE

Journal Requirements:

Reviewers' comments:

Reviewer's Responses to Questions

**Comments to the Author**

1. Is the manuscript technically sound, and do the data support the conclusions?

Reviewer #1: Yes

Reviewer #2: Yes

Reviewer #3: No

2. Has the statistical analysis been performed appropriately and rigorously? 

Reviewer #1: N/A

Reviewer #2: Yes

Reviewer #3: No

3. Have the authors made all data underlying the findings in their manuscript fully available?

Reviewer #1: Yes

Reviewer #2: Yes

Reviewer #3: No

4. Is the manuscript presented in an intelligible fashion and written in standard English?

Reviewer #1: Yes

Reviewer #2: Yes

Reviewer #3: Yes

5. Review Comments to the Author

Reviewer #1: This paper addresses an interesting issue (or set of them) through a set of semi-structured interviews applied to experts in the field. There are some comments that need your attention, as you will see below:

- Abstract could be structured and clearer, therefore.

- The paper is written in a reader friendly way. However, many typos are spread throughout the manuscript.

- Introduction is not really convincing, as literature review is weak and literature on behavioral contributors to usar safety seems absent from this paper, even though good points in this regard are made in the results.

- The methodological approach is superfluous. Qualitative research does not mean "technically poor" or "less rigorous" research. Please elaborate better on the methods used and the data analysis strategy.

- On the other hand, the results are more organized an present acceptable ledvels of detail, even in absence of the methodological setting of the study in the previous section.

- Discussion does not really discuss key findings, but re-describes and paraphrases most of them. This needs strong revisions from the authors.

- One of my main concerns is the lack of adequacy of the conclusions. As this is a very limited study in terms of coverage and data, my feeling is that what the authors conclude exceeds the actual scope of the data. Therefore, it would be suggestible to carefully revise each one of the conclusive statements, reshaping them if needed, in view of the key limitations of the study.

For instance, authors did well saying that their results "highlight" some expert perceptions. However, three lines below the statement is "data showed public transport modes have relatively lower risk of being seriously injured in road crash"... and this was not data directly related to your study, but it looks like anyway.

- Line 437: This sentence is incomplete.

Reviewer #2: The research aimed to analyse the role of modal shift of road users in improving road safety through semi-structured interviews with experts. This is an interesting topic with important practical applications. However, I consider that some modifications should be made to the manuscript before it is considered for publication.

The introduction should be developed further, setting out in detail the factors involved in modal choice and the implications for mobility and road safety. For example, evidence suggests that perceived safety may be a variable influencing route selection or the choice of a particular mode of transport for travel (e.g. doi: https://doi.org/10.3389/frsc.2020.00042 and https://doi.org/10.1016/j.tranpol.2012.09.009).

Additionally, I recommend including the main hypotheses of the study together with the research objectives.

The methodology and results are very comprehensive. And the discussion is adequate, having contrasted the results with similar research to explain and contextualise the data obtained. I only suggest including measures that can enhance modal shift based on the results and scientific evidence. The development of specific communication campaigns can be an effective tool for this purpose, especially if they are presented together with other complementary measures (e.g. doi: (https://doi.org/10.3390/safety7040066 and https://doi.org/10.5114/jhi.2016.61421).

Reviewer #3: The author failed to illustrate the conclusions with solid data and rigrous logitistics. The sample size of participants of interview is too small to obtain reliable conclusions. The author should expand the sample number and discuss the representativity of the samples.

Ambiguous or uncertain words are used in the results and conclusion parts, such as "somehow", "something" and so on.

The author should use more plots to directly show the results. For example, the percentages of each point of view.

6. PLOS authors have the option to publish the peer review history of their article (what does this mean?). If published, this will include your full peer review and any attached files.

Reviewer #1: No

Reviewer #2: No

Reviewer #3: No

---

## [Author Response · Author response to Decision Letter 0]

14 Dec 2022

General Comments Response

We followed PLOS ONE's style requirements in the revised manuscript. 

2. In your Data Availability statement, you have not specified where the minimal data set underlying the results described in your manuscript can be found. PLOS defines a study's minimal data set as the underlying data used to reach the conclusions drawn in the manuscript and any additional data required to replicate the reported study findings in their entirety. All PLOS journals require that the minimal data set be made fully available. For more information about our data policy, please see http://journals.plos.org/plosone/s/data-availability Interview manuscripts will be attached 

Interview manuscripts will be attached after accepting the paper 

Important: If there are ethical or legal restrictions to sharing your data publicly, please explain these restrictions in detail. Please see our guidelines for more information on what we consider unacceptable restrictions to publicly sharing data: http://journals.plos.org/plosone/s/data-availability#loc-unacceptable-data-access-restrictions. Note that it is not acceptable for the authors to be the sole named individuals responsible for ensuring data access. Interview manuscripts will be attached

Reviewer 1 

This paper addresses an interesting issue (or set of them) through a set of semi-structured interviews applied to experts in the field. There are some comments that need your attention, as you will see below:

Comment Response to Reviewer

- Abstract could be structured and clearer, therefore. The abstract was revised with structure subtitles and some rephrasing. 

- The paper is written in a reader friendly way. However, many typos are spread throughout the manuscript. The paper was revised and corrected 

Introduction is not really convincing, as literature review is weak and literature on behavioral contributors to usar safety seems absent from this paper, even though good points in this regard are made in the results. The introduction is revised, and a new section added about modal choice (travel behaviour) impact on road safety 

The methodological approach is superfluous. Qualitative research does not mean "technically poor" or "less rigorous" research. Please elaborate better on the methods used and the data analysis strategy. On the other hand, the results are more organized an present acceptable ledvels of detail, even in absence of the methodological setting of the study in the previous section. Section 2 includes the method it includes sub summary of the method steps in participant selection and the thematic analysis approach. The method section was revised to provide more clarity.

- Discussion does not really discuss key findings, but re-describes and paraphrases most of them. This needs strong revisions from the authors. The discussion section was revised and reproduced 

One of my main concerns is the lack of adequacy of the conclusions. As this is a very limited study in terms of coverage and data, my feeling is that what the authors conclude exceeds the actual scope of the data. Therefore, it would be suggestible to carefully revise each one of the conclusive statements, reshaping them if needed, in view of the key limitations of the study.

For instance, authors did well saying that their results "highlight" some expert perceptions. However, three lines below the statement is "data showed public transport modes have relatively lower risk of being seriously injured in road crash"... and this was not data directly related to your study, but it looks like anyway. The conclusion section was revised

Line 437: This sentence is incomplete.

 This is reproduced now 

Reviewer #2 

The research aimed to analyse the role of modal shift of road users in improving road safety through semi-structured interviews with experts. This is an interesting topic with important practical applications. However, I consider that some modifications should be made to the manuscript before it is considered for publication. The manuscript has been revised and modified 

consider that some modifications should be made to the manuscript before it is considered for publication.

The introduction should be developed further, setting out in detail the factors involved in modal choice and the implications for mobility and road safety. For example, evidence suggests that perceived safety may be a variable influencing route selection or the choice of a particular mode of transport for travel (e.g. doi: https://doi.org/10.3389/frsc.2020.00042 and https://doi.org/10.1016/j.tranpol.2012.09.009). The issue of perceived safety as a factor in modal choice was highlighted in the results and discussions where it might play a barrier to choosing cycling as it is perceived as a high-risk mode. It was also captured in the results and discussions around older people’s perception of risk from other public transport users’ behaviour which might influence their choice to use public transport. 

Thanks for the two references, and they were added to the discussion section. 

Additionally, I recommend including the main hypotheses of the study together with the research objectives. Thanks, it was added to the study objectives 

The methodology and results are very comprehensive. And the discussion is adequate, having contrasted the results with similar research to explain and contextualise the data obtained. Thanks, the manuscript was revised with further improvements 

I only suggest including measures that can enhance modal shift based on the results and scientific evidence This is beyond the scope of this study as it was only meant to explore if safety can encourage modal shift. However, we touched on that in the conclusion: “Working in parallel by addressing those factors and increasing the awareness of the different risks associated with different modes could potentially encourage the choice of more desirable safe modes”

The development of specific communication campaigns can be an effective tool for this purpose, especially if they are presented together with other complementary measures (e.g. doi: (https://doi.org/10.3390/safety7040066 and https://doi.org/10.5114/jhi.2016.61421) Thanks, communication and information availability were added to the discussion and the two references were added 

Reviewer #3:

The author failed to illustrate the conclusions with solid data and rigrous logitistics. The sample size of participants of interview is too small to obtain reliable conclusions. The author should expand the sample number and discuss the representativity of the samples. The manuscript and conclusions were revised to better illustrate the study. 

The sample size is enough as suggested by previous qualitative research, and this was discussed in the method section and supported by the previous literature. 

Ambiguous or uncertain words are used in the results and conclusion parts, such as "somehow", "something" and so on.

The author should use more plots to directly show the results. For example, the percentages of each point of view. The manuscript and conclusions were revised to better illustrate the study.

We don’t think using percentages for this qualitative study is necessary as the thematic analysis uses saturation as an approach to describe the bulk of ideas.

---

## [Decision Letter · Decision Letter 1]

12 Jan 2023

The role of safety in modal shift choice and safety: A transport expert perspective in the state of Victoria (Australia)

PONE-D-22-23849R1

Dear Dr. Mohammad Nabil Ibrahim

We’re pleased to inform you that your manuscript has been judged scientifically suitable for publication and will be formally accepted for publication once it meets all outstanding technical requirements.

Kind regards,

Iman Aghayan, Ph.D.

Academic Editor

PLOS ONE

Additional Editor Comments (optional):

Reviewers' comments:

Reviewer's Responses to Questions

**Comments to the Author**

1. If the authors have adequately addressed your comments raised in a previous round of review and you feel that this manuscript is now acceptable for publication, you may indicate that here to bypass the “Comments to the Author” section, enter your conflict of interest statement in the “Confidential to Editor” section, and submit your "Accept" recommendation.

Reviewer #2: All comments have been addressed

Reviewer #3: All comments have been addressed

2. Is the manuscript technically sound, and do the data support the conclusions?

Reviewer #2: Yes

Reviewer #3: Yes

3. Has the statistical analysis been performed appropriately and rigorously? 

Reviewer #2: Yes

Reviewer #3: Yes

4. Have the authors made all data underlying the findings in their manuscript fully available?

Reviewer #2: Yes

Reviewer #3: Yes

5. Is the manuscript presented in an intelligible fashion and written in standard English?

Reviewer #2: Yes

Reviewer #3: Yes

6. Review Comments to the Author

Reviewer #2: The authors have taken into account the suggestions I provided in my previous review, so I consider that the manuscript is suitable for publication.

Reviewer #3: The authors have made revision according to the suggestions from the reviewers. I recommend this manuscript being accepted

7. PLOS authors have the option to publish the peer review history of their article (what does this mean?). If published, this will include your full peer review and any attached files.

Reviewer #2: No

Reviewer #3: No

---

## [Editor Report · Acceptance letter]

27 Feb 2023

PONE-D-22-23849R1 

The role of safety in modal choice and shift: A transport expert perspective in the state of Victoria (Australia) 

Dear Dr. Ibrahim:

I'm pleased to inform you that your manuscript has been deemed suitable for publication in PLOS ONE. Congratulations! Your manuscript is now with our production department. 

Kind regards, 

on behalf of

Dr. Iman Aghayan 

Academic Editor

PLOS ONE